# Early Detection of Diabetic Peripheral Neuropathy by fMRI: An Evidence-Based Review

**DOI:** 10.3390/brainsci12050557

**Published:** 2022-04-26

**Authors:** Ahish Chitneni, Adam Rupp, Joe Ghorayeb, Alaa Abd-Elsayed

**Affiliations:** 1Department of Rehabilitation and Regenerative Medicine, NewYork-Presbyterian Hospital—Columbia and Cornell, New York, NY 10065, USA; 2Department of Physical Medicine and Rehabilitation, University of Kansas Health System, Kansas City, MO 66160, USA; arupp01@atsu.edu; 3Department of Physical Medicine and Rehabilitation, University of Medicine & Health Sciences, New York, NY 10001, USA; joe.ghorayeb@gmail.com; 4Department of Anesthesia, Division of Pain Medicine, School of Medicine & Public Health, University of Wisconsin, Madison, WI 53726, USA

**Keywords:** functional magnetic resonance imaging, diabetes, neuropathic pain, diabetic peripheral neuropathy

## Abstract

With the significant rise in the prevalence of diabetes worldwide, diabetic peripheral neuropathy (DPN) remains the most common complication among type 1 and 2 diabetics. The adverse sequelae of DPN, which include neuropathic pain, diabetic foot ulcers and lower-limb amputations, significantly impact quality of life and are major contributors to the biopsychosocial and economic burden of diabetes at the individual, societal and health system levels. Because DPN is often diagnosed in the late stages of disease progression by electromyography (EMG), and neuropathic pain as a result of DPN is difficult to treat, the need for earlier detection is crucial to better ascertain and manage the condition. Among the various modalities available to aid in the early detection of DPN, functional magnetic resonance imaging (fMRI) has emerged as a practical tool in DPN imaging due to its noninvasive radiation-free nature and its ability to relate real-time functional changes reflecting the local oxygen consumption of regions of the CNS due to external stimuli. This review aims to summarize the current body of knowledge regarding the utility of fMRI in detecting DPN by observing central nervous system (CNS) activity changes among individuals with DPN when compared to controls. The evidence to date points toward a tendency for increased activity in various central neuroanatomical structures that can be detected by fMRI and positively correlates with diabetic neuropathic pain.

## 1. Introduction

The International Diabetes Federation (IDF) predicts the global prevalence of diabetes mellitus (DM) to rise to 643 million people by 2030 and 783 million by 2045 [1]. Diabetic peripheral neuropathy (DPN) is a common late complication of diabetes mellitus (DM) that affects up to 50% of DM patients and causes progressive disability [2]. DPN is described as a symmetrical, length-dependent sensorimotor polyneuropathy caused by metabolic changes and microvascular alterations resulting from exposure to hyperglycemia and other associated risk factors [2]. In the United States, the annual cost for managing painful DPN and foot ulceration with lower limb amputation has been estimated to be between USD 4.6–13.7 billion [3]. Neuropathic pain is a common symptom, affecting approximately 25% of those with DPN [4], and patients with neuropathic pain often respond poorly to analgesic medications [5]. Among individuals with DM and DPN, those with painful DPN incur baseline costs that are 20% higher than diabetic controls and are 200%, 356%, and 224% more likely to use opioids, anticonvulsants, and antidepressants, respectively, when compared with diabetic controls [6,7]. Treatment of neuropathic pain is extremely difficult, with no true disease-modifying therapies for most underlying conditions. Part of the challenge of drug development and clinical trial design in this respect is the lack of robust biomarkers for neuropathy and neuropathic pain [5]. DPN is often thought to be a disease of the peripheral nervous system. However, some evidence suggests that CNS changes may also account for disease pathogenesis [8,9,10,11]. The current gold-standard method of diagnosing DPN is electromyography (EMG), though many DPN patients do not present with neuropathic symptoms despite showing evidence of defective nerve function on EMG. As a result of the insidiousness of onset of symptoms, DPN is often not diagnosed in the early stages. Evaluating patients with neuropathy involves a detailed history and physical examination including a review of current and past medications. Although there are no standard laboratory or imaging studies to test for peripheral neuropathies, clinicians often utilize various studies to aid in the diagnosis and help narrow down the underlying cause of the neuropathy whether it be inflammatory, infectious or metabolic, including but not limited to: HbA1c testing, testing for vitamin and mineral deficiencies, metabolic panels and heavy metal toxicities [12], infectious workup [13,14], thyroid function tests [15], antibody testing for specific autoimmune diseases known to cause peripheral neuropathy [16], and nerve biopsy [17]. Among the various modalities available to aid in the early detection of DPN, fMRI has emerged as a practical tool in DPN imaging due to its noninvasive radiation-free nature and its ability to relate real-time functional changes reflecting the local oxygen consumption of regions of the CNS due to external stimuli [18]. fMRI is a method of detecting dynamic patterns of activity in the working human brain through the depiction of changes in deoxyhemoglobin concentration consequent to task-induced or spontaneous modulation of neural metabolism as well as blood flow increases compared to oxygen metabolism when local neural activity increases. The effects noted translate to subtle increases in the local magnetic resonance signal and the blood-oxygenation-level-dependent (BOLD) effect when neural activity increases [19]. In addition to detecting BOLD responses from activity due to tasks or stimuli, fMRI can also measure resting or negative-task state, which shows individual baseline BOLD variance [20]. In the clinical context, fMRI has demonstrated utility in the assessment of various diseases and disorders such as Alzheimer’s disease [21], Parkinson’s disease [22], autism [23], schizophrenia [24], and epilepsy [25]. The aim of this narrative review is to summarize the evidence to date regarding the ability of fMRI to detect the early stages of CNS impairment caused by DPN.

## 2. Materials and Methods

### 2.1. Search Strategy

A MEDLINE/PubMed search using the terms “functional magnetic resonance imaging” and “diabetic neuropathy” and “pain” for entries between 1 January 1988 and 24 March 2022 was conducted. The initial search yielded 91 entries. Studies were included if they met the following criteria: (1) published in English and in a peer-reviewed journal; (2) study designs included experimental (Randomized Clinical Trials) and observational (Cohort and Case–Control) studies; and (3) study populations were limited to adults (19+ years old). Case reports, opinions, comments, letters to the editor, and articles without scientific data or a report of their methodology were excluded. Authors AC and AR conducted the study acceptance and rejection process with the author JG acting as a tie-breaker where necessary. After the exclusion and inclusion process, a total of 9 papers were selected for review.

### 2.2. Level of Evidence

A level of evidence rating was applied to every study based on the criteria outlined in (Table 1). Randomized double-blinded comparative clinical research of good quality and efficient size obtained a level of evidence A2, while cohort studies not meeting these criteria or case-control studies obtained a level of evidence B. Non-controlled trials obtained a level of evidence C.

### 2.3. Strength of Conclusion

The strength of conclusion (ranging from 1 to 4) was calculated for each cluster of studies reflecting one outcome parameter (Table 2) and is denoted under Table 3, which summarizes the findings of the studies reviewed reflecting the effectiveness of fMRI at detecting painful diabetic neuropathy. Strength of conclusion 1 was assigned for a study of level A1 or at least 2 independently conducted studies of level A2. Strength of conclusion 2 was given when at least 2 independently conducted studies of evidence level B or one trial of evidence level A2 was included in the cluster, and strength of conclusion 3 was assigned if one study of evidence level B or C was present. Strength of conclusion 4 was given in case of inconclusive or inconsistent results between various studies.

## 3. Results

Of the nine studies included in the review, eight studies demonstrated significant fMRI changes in individuals with DPN, demonstrating a positive correlation with the condition, while one study did not report any significant CNS differences between individuals with DPN and healthy controls.

## 4. Discussion

Teh and colleagues sought to stratify clinical pain phenotypes of DPN by way of an observational cohort study of 43 individuals with painful DPN and discerning cortical functional connectivity between phenotypes by resting-state fMRI (RS-fMRI) [26]. The study participants consisted of right-hand dominant adults with painful DPN for at least 6 months. Thirty-three of the study participants were identified as presenting with the irritable clinical phenotype (IR), defined as relatively preserved sensory function associated with thermal and/or mechanical hyperalgesia, and the remaining 10 study participants were identified as presenting with the non-irritable clinical phenotype (NIR), defined as presenting as insensate to nociceptive stimuli. The main findings from this study revealed that individuals with the IR nociceptor phenotype had significantly greater thalamus–insular cortex functional connectivity and decreased thalamus–somatosensory cortex functional connectivity compared with those with the NIR nociceptor phenotype. A positive correlation was also noted between thalamus–insular cortex functional connectivity and pain scores. This inference suggests that the insular cortex, which is known to play a pivotal role in affective and attentional pain processing, may be an overactive pain-promoting brain region in individuals with the IR nociceptor phenotype. Additionally, a greater reduction in thalamus–somatosensory cortex functional connectivity in individuals with more severe neuropathy was appreciated. This relationship suggests that the deafferentation resulting from a severe neuropathy leads to reduced peripheral sensory input, which, in turn, leads to a reduction in somatosensory cortical volume and functional connectivity [26].

A cross-sectional case-control study conducted by Li et al. was conducted to explore MRI changes in response to thermal stimuli in patients with diabetes and comparing the findings against a control group [27]. In the study, a total of 36 participants were enrolled: 8 patients with T2DM and diabetic peripheral neuropathy, 13 patients with T2DM without diabetic peripheral neuropathy, and 15 participants with no diagnosis of T2DM or symptoms of DPN. All study participants underwent thermal stimuli and were asked to rate their level of pain and itch perception on a VAS. fMRI imaging was analyzed and brain imaging changes were recorded. Compared to the healthy group, imaging for the DPN group showed increased activation of all areas of the brain, including the caudate nucleus, frontal gyrus, temporal lobes, and hippocampus. Overall, given the fMRI changes seen, the study authors note that fMRI may be useful in the detection of nervous system impairment due to diabetic peripheral neuropathy [27].

Zhang et al. conducted a cross-sectional case–control study to determine whether individuals with DPN demonstrate structural changes in areas of the brain that are associated with proprioception, touch, pain and temperature perception, in addition to motor functions [28]. The study authors enrolled 67 patients with T2DM; 44 patients had non-painful DPN, and 23 patients presented with painful DPN lasting at least 6 months, who were compared to 88 age- and sex-matched healthy controls (HCs). Surface-based morphometry (SBM), a technique used to construct and analyze brain surfaces in order to delineate structural boundaries, was utilized to detect structural differences between groups. Diffusion parameter analysis was also employed to detect the presence of white matter changes. Individuals with both painful and non-painful DPN showed thinner cortical structures and greater cortical surface area activation when compared to healthy controls (*p* < 0.05), with no meaningful differences noted between DPN groups (16). The remainder of the study findings support the notion that CNS alterations contribute to DPN disease progression by way of both gray and white matter changes in specific brain regions [28].

Another study by Selvarajah et al. evaluated the use of fMRI to examine somatosensory cortex changes in patients with diabetic peripheral neuropathy [29]. In this study, 44 patients with a T1DM diagnosis of greater than 5 years, aged 18–65 years, and with a hemoglobin A1C level of <11% were enrolled in the study. Results showed that patients with painful diabetic peripheral neuropathy were found to have a lower somatosensory cortical thickness. Additionally, the severity of peripheral neuropathy correlated with the anatomical changes seen in fMRI. One of the important findings of the study is that alterations in the somatosensory cortex are related to the severity and presence of diabetic peripheral neuropathy symptoms [29].

In another cross-sectional study by Zhang et al., 37 patients (19 with PDN and 18 patients with non-pain neuropathy) and 15 gender- and age-matched HCs underwent blood-oxygenation-level-dependent (BOLD) imaging and rs-fMRI to characterize brain activity, and nerve and mental scale assessments (serving as potential clinical correlates of neuropsychiatric function) to discern correlations between brain activities and clinical indicators that may provide clues for the diagnosis and treatment of painful diabetic neuropathy (PDN) [30]. The authors noted that patients with PDN had increased insulin resistance (*p* = 0.03), increased depression (*p* = 0.02) and increased anxiety (*p* < 0.001) compared with the HCs [30]. All of these conditions were associated with abnormal spontaneous activities in several regions of the brain, including the somatosensory, cognitive and emotional regions. The authors did note the lack of a diabetic control group without neuropathy, the small study sample size and study design, and the lack of diabetes-related cognitive memory function assessment measures as limitations.

In a study by Hansen et al., 48 participants with a diagnosis of type 1 diabetes mellitus (T1DM) and evidence of diabetic peripheral neuropathy and 28 participants with no diagnosis or symptoms participated in a cross-sectional case–control study to evaluate findings on brain imaging [31]. After imaging was conducted and analyzed, the patients in the diabetes control had a significant reduction in gray matter volume and bilateral thalamus volume compared to the control group. In this group, the thalamic volume was associated with intra-thalamic NAA/cre levels but did not have any significant decrease based on the duration of diabetes, severity of neuropathic pain or the presence of pain. Overall, this study provides some insight into the understanding of the pathophysiology of the perception changes seen in patients with diabetic peripheral neuropathic pain [31].

Segerdahl et al. conducted a case–control study involving 30 subjects [32]. Neuropathy was confirmed via physical exam, nerve conduction studies, skin biopsy and serial questionnaires. The group was then divided based on painful and painless DPN. The authors utilized fMRI to evaluate pathways involving the ventrolateral periaqueductal gray (vlPAG). They found increased connectivity between the vlPAG and the cortex, which correlated with those patients who reported increased thermal hyperalgesia (burning pain). This positive relationship was correlated for the painful DPN group but not the no-painful group (*p* < 0.01) [32].

Tseng et al. conducted a case–control study involving 33 study participants (n = 11 per group) with painful DPN, painless DPN and healthy controls [33]. The authors utilized fMRI while applying a 44 °C heat stimuli to the right foot to induce neuropathic pain symptoms. They found augmented responses in the limbic and striatal areas. Blood oxygen signals were positively correlated with pain ratings to stimulations in the painful group, unlike the non-painful group who reported reductions in pain ratings. The authors suggest that enhanced limbic and striatal activations could underlie maladaptive responses and contribute to the burning sensations experienced by individuals with painful DPN [33].

Cauda et al. conducted a case–control study on thalamocortical functional connectivity in eight patients with DPN and compared it with eight healthy subjects [34]. Enrolled patients had pain for >2 years and had a 1-month medication washout prior to imaging. fMRI evaluations of the primary somatosensory cortex, ventral posterior lateral (VPL) thalamic nucleus and medial dorsal (MD) thalamic nucleus showed decreased resting state functional connectivity compared to the control group. The authors concluded that chronic pain alters thalamocortical connections, causing a disruption of thalamic feedback and perceptions of pain (allodynia and hyperalgesia) in addition to a disruption in the modulation of emotional responses to pain [34].

## 5. Limitations

Although a comprehensive summary regarding the efficacy of fMRI at detecting DPN was attempted, a primary limitation of this review includes finite access to data, hence the use of a single search database. Further limitations include the available study designs reviewed (e.g., cohort, case–control and cross-sectional studies), which include small samples of study participants. Future large-scale randomized clinical trials comparing individuals with painful DPN and painless DPN, as well as individuals with diabetes but without DPN, in addition to healthy controls, would help to further and more confidently ascertain fMRI’s role in early detection of DPN.

## 6. Conclusions

Diabetic peripheral neuropathy, specifically the painful variant, is a very common and debilitating condition. This focused review on the detection of painful DPN by fMRI brings to light the changes in functional activity and blood flow in the central nervous system, which may provide clinicians an avenue for early detection and management before the late stages of disease pathology are realized. The studies reviewed demonstrate a positive correlation between increased activity in both the gray and white matter structures in the brain and neuropathic pain symptoms. Our review highlights the importance of ongoing investigations in order to further elucidate the pathophysiological processes that underlie painful diabetic peripheral neuropathy in an attempt to intercept and halt or reduce further disease progression, ultimately improving clinical outcomes.

## Figures and Tables

**Table 1 brainsci-12-00557-t001:** Level of Evidence.

A1	Systematic Review of ≥2 A2-Level Studies
A2	Randomized double-blinded clinical trial of good quality and adequate size
B	Comparative/controlled studies failing to satisfy criteria for A2
C	Non-comparative studies
D	Expert opinion

**Table 2 brainsci-12-00557-t002:** Strength of Conclusion.

Level	Conclusion Based on
1	A1 study or ≥2 A2-level studies
2	One A2-level study or ≥2 independent B-level studies
3	One B-level or C-level study
4	Inconclusive or inconsistent results between various studies

**Table 3 brainsci-12-00557-t003:** Summary of findings from included studies suggesting fMRI is effective at detecting painful diabetic neuropathy.

Author	Year	Study	Conclusions	Level of Evidence
Cauda	2009	Case–control	Chronic pain decreases thalamocortical connections causing disruptions in pain perception and emotional responses related to pain.	B
Segerdahl	2018	Case–control	Painful neuropathic pain was found to have a statistically significant positive correlation with increased ventrolateral periaqueductal gray connections (*p* < 0.01) relative to non-painful DPN—possibly relating to increased hyperalgesia and allodynia.	B
Tseng	2012	Case–control	A positive correlation was found between painful DPN and activation in the limbic and striatal areas. This enhanced activity could underlie the burning pain sensations experienced by individuals with painful DPN.	B
Hansen	2021	Cross-sectional case–control	Thalamic volume was associated with intra-thalamic NAA/cre levels but did not have any significant decrease based on the duration of diabetes, severity of neuropathic pain, or the presence of pain.	B
Li	2018	Cross-sectional case–control	In the DPN group, compared to the healthy group, imaging showed an increased activation of all areas of the brain including the caudate nucleus, frontal gyrus, temporal lobes, and hippocampus.	B
Selvarajah	2019	Cross-sectional	Patients with painful diabetic peripheral neuropathy were found to have a lower somatosensory cortical thickness and the severity of DPN was correlated with the anatomical changes seen on fMRI imaging.	C
Teh	2021	Cohort study	A positive correlation was also noted between thalamus–insular cortex functional connectivity and pain scores. Additionally, a greater reduction in thalamus–somatosensory cortex functional connectivity in individuals with more severe neuropathy was appreciated—suggesting that the deafferentation resulting from severe neuropathy leads to a reduction in somatosensory cortical volume and functional connectivity.	C
Zhang	2019	Cross-sectional case–control	Patients with PDN had increased insulin resistance (*p* = 0.03), increased depression (*p* = 0.02) and increased anxiety (*p* < 0.001) compared with the HCs. All of these conditions were associated with abnormal spontaneous activities in several regions of the brain, including the somatosensory, cognitive and emotional regions.	B
Zhang	2020	Cross-sectional case–control	Significant GM and WM alterations in some key brain regions of the ascending spinal–cortical somatosensory pathway, the descending motor pathway, and pain perception and modulation in patients with DPN was appreciated.	B

Abbreviations: fMRI: functional magnetic resonance imaging, DPN: Diabetic peripheral neuropathy, T1DM: Type 1 diabetes mellitus, T2DM: Type 2 diabetes mellitus, NAA/cre: N-acetylaspartate/creatine, RS-fMRI: resting state fMRI, HCs: Healthy controls, GM: Gray matter, WM: White matter, PDN: Painful diabetic neuropathy. Strength of Conclusion: 2.

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
