# Peer review of "Early Detection of Diabetic Peripheral Neuropathy by fMRI: An Evidence-Based Review"

_brainsci, 2022, doi:10.3390/brainsci12050557_

Round 1

Reviewer 1 Report

Comments:

  1. The title is interesting and well written by the authors but still, lacks some issues like the abstract section is poorly written. The authors are advised to add more information that will allow a reader to understand the key findings of the review.
  2. Kindly remove fMRI from your keywords section as its full form is also there.
  3. The introduction part of the manuscript is not written well which constitutes the lack of crux in it. The major part of the introduction consists of diabetic neuropathy and is less focused on the imaging techniques for the detection of neuropathy.
  4. A kind of advice to the authors on why they only stick to ‎PubMed they to search on other databases like ‎EMBASE, ‎Cochrane Library, UpToDate, and PubMed Central (PMC)
  5. What is the duration of study or data collection?
  6. The manuscript needs standard review.
  7. Please do add the future perspective of this study.
  8. Mention search strategy, inclusion, and exclusion criteria in the manuscript.

Author Response

Reviewer 1:

  1.   The title is interesting and well written by the authors but still, lacks some issues like the abstract section is poorly written. The authors are advised to add more information that will allow a reader to understand the key findings of the review.

Thank you for your thoughtful review and insights. Please see attached our revised manuscript with our enhanced abstract providing more information for your review and consideration.

  1.     Kindly remove fMRI from your keywords section as its full form is also there.

Done.

  1.   The introduction part of the manuscript is not written well which constitutes the lack of crux in it. The major part of the introduction consists of diabetic neuropathy and is less focused on the imaging techniques for the detection of neuropathy.

Thank you for highlighting this for us. We have amended the introduction to better reflect the crux of our manuscript for your review and consideration.

  1.   A kind of advice to the authors on why they only stick to ‎PubMed they to search on other databases like ‎EMBASE, ‎Cochrane Library, UpToDate, and PubMed Central (PMC)

Thank you for this insight. We agree with this recommendation. We included an explanation in the “Limitations” section of the revised manuscript highlighting that finite access to data was a main limitation of this review, hence our use of PubMed as our single search engine.

  1.     What is the duration of study or data collection?

 We have included the custom range in our methods beginning at January 1, 1988 to March 24, 2022.

  1.     The manuscript needs standard review.

We don’t know what this means. Further clarification in this regard will be helpful to make further revisions where necessary.

  1.   Please do add the future perspective of this study.

 Thank you. We have added this information in our “Limitations” section in the revised manuscript.

  1.     Mention search strategy, inclusion, and exclusion criteria in the manuscript.

Thank you. We have included this information in our revised draft for your review and consideration. We hope that our revised manuscript now merits publication in Brain Sciences.

Reviewer 2 Report

Minor Revision:

  1. Introduction- Authors should also write about the importance of fMRI, if possible with some examples (with citations) where fMRI has already proved to be an important tool to detect early onset of any other disease (s). 
  2. Authors should elaborate methods a little more, especially emphasizing why only 9 papers were accepted for review. Authors may write one or two very strong points which is show-cased in all the 9 papers.
  3. In table 1, also should provide one more column to write about strength of each study, or if possible weaknesses also. 
  4.  Some of the citations are very old, for example Gordois et al., reported healthcare costs DPN in US in 2003. Similarly there are other old citations which should be replaced by newer one. 

Author Response

Reviewer 2:

  1.   Introduction- Authors should also write about the importance of fMRI, if possible with some examples (with citations) where fMRI has already proved to be an important tool to detect early onset of any other disease (s). 

Thank you for your thoughtful review and insights. We have revised our introduction considerably to include the aforementioned recommendations for your review and consideration.

  1.   Authors should elaborate methods a little more, especially emphasizing why only 9 papers were accepted for review. Authors may write one or two very strong points which is show-cased in all the 9 papers.

Thank you. We have elaborated further in the methods section of our revised draft for your review and consideration, and we have also included two tables to apply levels of evidence and strength of study conclusions to the studies we reviewed. 

  1.     In table 1, also should provide one more column to write about strength of each study, or if possible weaknesses also. 

Thank you for this recommendation. We have applied a level of evidence rating and strength of conclusion rating as well. The level of evidence is included in an additional column with the corresponding study and the strength of conclusion rating is provided underneath Table 3. We hope this additional information will satisfy the requirements of this recommendation. 

  1.     Some of the citations are very old, for example Gordois et al., reported healthcare costs DPN in US in 2003. Similarly there are other old citations which should be replaced by newer one. 

Thank you for this recommendation. We have replaced older citations with newer citations to better reflect the information we wish to convey in a more temporal fashion (specifically as it relates to painful DPN). At this time, the only citation we could locate that specifically describes the healthcare cost of DPN in the U.S. is the Gordois et al. study. We hope our revisions in this regard meet the requirements for the aforementioned recommendations and that our revised manuscript now merits publication in Brain Sciences.